# Effect of DA-9701 on the Gastrointestinal Motility in the Streptozotocin-Induced Diabetic Mice

**DOI:** 10.3390/jcm10225282

**Published:** 2021-11-13

**Authors:** Changyoon Ha, Heejin Kim, Rari Cha, Jaemin Lee, Sangsoo Lee, Jung-Hwa Ryu, Hyunjin Kim, Ok-Jae Lee

**Affiliations:** 1Department of Internal Medicine, Gyeongsang National University College of Medicine, Jinju 52727, Korea; cyha@gnu.ac.kr (C.H.); heejin1231@hanmail.net (H.K.); rari83@naver.com (R.C.); 01179jm@naver.com (J.L.); 3939lee@naver.com (S.L.); 2Division of Gastroenterology, Gyeongsang National University Hospital, Jinju 52727, Korea; 3Institute of Health Sciences, Gyeongsang National University, Jinju 52727, Korea; 4Division of Gastroenterology, Gyeongsang National University Changwon Hospital, Changwon 51472, Korea; 5Department of Internal Medicine, Ewha Womans University College of Medicine, Seoul 07804, Korea; singrum@gmail.com

**Keywords:** DA-9701, diabetic mouse model, functional dyspepsia, diabetic gastroparesis, STZ, gastrointestinal motility

## Abstract

Background: Compared to the general population, diabetic patients experience more frequent episodes of gastrointestinal (GI) motility dysfunction, owing to the disruption of functional innervations. DA-9701 is a new prokinetic agent formulated from the extracts of *Pharbitidis* semen and *Corydalis* tuber. Aim: To investigate the effect of DA-9701 on GI motility in an animal model of streptozotocin (STZ)-induced diabetes. Methods: Diabetes was induced in mice by intraperitoneal injection of STZ (40 mg/kg of body weight in 0.1 M citrate buffer) for 3 days. Diabetic mice were divided into four groups and administered DA-9701 in different doses (1, 3, and 10 mg/kg) or placebo for 2 weeks. Intestinal transit was assessed using charcoal meal movement. GI isometric contraction was measured by applying an isometric force transducer on a circular muscle strip of the antrum, ileum, and proximal colon of sacrificed mice. Gastric emptying rate was evaluated by measuring the dye percentage remaining in the stomach relative to the total dye amount recovered in a standardization group of mice. Results: Body weight and antral and small intestinal motility were less in diabetic mice than in control mice, and colonic motility was similar in both. DA-9701 showed a dose-dependent increase in the amplitude of spontaneous phasic contractions in the antrum, ileum, and colon in diabetic mice without influencing body weight or blood glucose levels. The degree of improvement was comparable between diabetic and control mice. Intestinal transit was significantly more delayed in diabetic mice than in controls (43 ± 7% vs. 67 ± 8%, *p* < 0.05); however, DA-9701 restored the delayed intestinal transit more effectively compared to placebo (75% vs. 50%). The gastric emptying rate was significantly more delayed in diabetic mice than in controls (43 ± 10% vs. 62 ± 12%, *p* < 0.05), and was improved by DA-9701 in a dose-dependent manner (50%, 55%, and 60% in mice treated with 1, 3, and 10 mg/kg of DA-9701, respectively, vs. 43% in placebo-treated and 60% in control mice). Conclusions: DA-9701 improved GI contractility without affecting blood sugar and body weight in diabetic mice. DA-9701 could improve the decreased GI motility and clinical symptoms in progressive diabetic patients.

## 1. Introduction

Functional gastrointestinal (GI) disorders occur more frequently in patients with diabetes than in the general population. Longstanding diabetes induces GI motility dysfunction via the disruption of nerve functions regulating the motility of the gut, which causes incomplete emptying of the different sections of the gastrointestinal tract. This process finally leads to gastroenteropathy, a composite disorder of the esophagus, stomach, small intestine, and colon [1]. Thus, diabetic patients experience delayed gastric emptying (GE) and various GI symptoms, such as nausea, vomiting, early satiety, bloating, postprandial discomfort, anorexia, weight loss, and abdominal pain. Recently, more concise pathophysiologic mechanisms of diabetic gastroparesis have been elucidated; these include extrinsic denervation of the stomach, causing delayed gastric emptying, and loss of nitric oxide synthase (NOS) in the enteric nerve, causing impaired inhibitory input, which induces decreased gastric accommodation and decreased gastric emptying, pylorospasm, and altered function of immune cells, such as type 2 macrophages, which can trigger damage to the interstitial cells of Cajal (ICC) and smooth muscle atrophy [2,3].

DA-9701 is a new prokinetic agent formulated from the extracts of *Pharbitidis* semen and *Corydalis* tuber [4]. This medicine, which is known to function as a 5-HT_1A_ agonist, 5-HT_4_ agonist, and 5-HT_3_ partial antagonist, was developed for the treatment of functional dyspepsia [4,5,6]. *Pharbitidis* semen is known to have natural ingredients that control edema, fullness, fecal and urinary retention, phlegm, fluid retention, and abdominal pain due to parasitic infestations. *Corydalis* tuber is known to be effective in adjusting mild depression, severe nerve damage, tremors, and intestinal spasm [7].

In this study, we investigated the effect of DA-9701 on gastrointestinal motility, the intestinal transit, and gastric emptying rate in a streptozotocin (STZ)-induced diabetic mouse model.

## 2. Materials and Methods

### 2.1. Streptozotocin (STZ)-Induced Diabetic Mice Model

All animal care and experimental procedures were approved by the Ethics of Animal Experiments Committee of Kyungpook National University. Six-week-old male Institute of Cancer Research (ICR) mice weighing 25–30 g were used as experimental animals. They were housed at 24 °C, were allowed free access to water and feed, and lighting was repeated in a darkness and lighting cycle every 12 h. After acclimatization for 1 week, the mice were randomly divided into two groups: the STZ-induced diabetic group and the normal control group.

STZ was dissolved in 0.1 mM citric acid buffer solution (STZ 40 mg/kg, pH 4.0) every day and injected via the intraperitoneal (IP) route for 3 consecutive days. In the normal control group, the same volume of citric acid buffer was injected intraperitoneally (Figure 1). Body weights were measured weekly during the experimental period. Two weeks after the first day of STZ injection, blood was collected from the tail vein and blood glucose levels were measured in each mouse. Individuals with a level of ≥300 mg/dL were considered to have diabetes. Blood glucose levels were measured at random times without overnight fasting using a GlucoDr^™^ Plus (Allmedicus, Anyang, Gyeonggi-do, Republic of Korea) test strip and a measuring instrument.

After the acclimatization phase of 7 days, streptozotocin was injected via intraperitoneal at a dose of 40 mg/kg for 3 days. The same volume of citric acid buffer was injected into normal control mice. Blood glucose level was measured after 14 days. Normal control and streptozotocin-induced mice were treated with DA-9701 in different doses (1, 3, and 10 mg/kg) for 14 days. On day 28, all mice were sacrificed, and gastrointestinal motility was assessed.

### 2.2. DA-9701 Treatment in Experimental Mice

The STZ-induced diabetic mice were divided into four groups, each with 10 mice: STZ-induced diabetic mice treated with different doses of DA-9701 at 1, 3, and 10 mg/kg, and STZ-induced diabetic mice with the placebo treatment. DA-9701 was suspended and administered orally at different doses of 1, 3, and 10 mg/kg for 2 weeks, and the placebo group was orally administered 3% hydroxypropyl methylcellulose (HPMC) for 2 weeks (Figure 1). To show the maximum changes in GI muscular contractility according to the DA-9701 dose escalation, the normal control or STZ-induced diabetic mice were also treated at higher doses (>10 mg/kg; 30, 100, and 300 mg/kg). In this experiment, the mice were exposed to 1, 3, 10, 30, 100, and 300 mg/kg of DA-9701.

### 2.3. Assessment of GI Motility Using Isometric Contraction Measurement

All studies were performed on normal or STZ mice after 2 weeks of treatment with DA-9701 or placebo. DA-9701 was given to the mice in different doses of 1, 3, 10, 30, 100, and 300 mg/kg. After 16 h of fasting, the mice were sacrificed. GI motility was evaluated by measuring isometric contractions in each segment of the bowel using circular muscle strips from the antrum, ileum, and proximal colon. From the sacrificed mice, 1–1.5 cm circular muscle strips of the antrum, ileum, and proximal colon were isolated and flushed with Krebs solution. The muscle strips were immediately placed in a 10 mL organ bath containing oxygenated (95% O_2_ + 5% CO_2_) Krebs solution at 37 °C. The distal end of the muscle segment was tied to a fixed mount and the tied proximal end was fixed to an isometric force-displacement transducer (FT-03, Grass-telefactor, Providence, RI, USA). Tension was monitored using an isometric force transducer and an index of the longitudinal muscle response was recorded. The signal was analyzed using a digital recording system. The signals from the transducers were processed using Powerlab 4/30 and Chart 7.2 (AD Instruments, Bella Vista, Australia).

The composition of Krebs solution was 10.1 mM glucose, 115.5 mM NaCl, 21.9 mM NaHCO_3_, 4.61 mM KCl, 1.14 mM NaH_2_PO_4_, 2.5 mM CaCl_2_, and 1.16 mM MgSO_4_.

### 2.4. Measurement of Intestinal Transit

All experiments were performed after 2 weeks of treatment with DA-9701 or 3% HPMC (placebo). After 16 h of fasting, the mice were administered a single dose of liquid charcoal meal (10% *w*/*v* charcoal suspension in 5% *w*/*v* suspension of acacia). Each subject was treated with a charcoal meal of 0.1 mL/10 g (body weight) via the oral route. All mice were euthanized via cervical dislocation 30 min after administration of the charcoal meal. To measure intestinal transit, the stomach and small intestine were isolated. They were then extended to a clean surface. The distance moved by the charcoal meal from the pylorus and the total length of the small intestine were measured. Intestinal transit was expressed as a percentage of the distance traveled by charcoal over the total length from the pylorus to the cecum.

### 2.5. Measurement of GE Rate with Phenol Red Marker

GE was determined by the phenol red method. All studies were performed on mice after 2 weeks of treatment with DA-9701 or 3% HPMC. After 16 h of fasting, the mice were treated orally with 1.5% carboxymethylcellulose (Sigma-Aldrich, Millipore, St. Louis, MI, USA) containing 0.05% phenol red (Sigma-Aldrich, Millipore, St. Louis, MI, USA) at a dose of 0.1 mL/10 g, 1 h after treatment with 3% HPMC or DA-9701. After another 30 min, the mice were sacrificed by cervical dislocation. The abdomen was opened carefully and the gastroesophageal junction and pylorus were tied to prevent the contents from flowing out; the stomach was then separated. The extracted stomach was ground using a homogenizer in 0.1 M NaOH, and the suspension was allowed to stabilize for 1 h at room temperature. Then, trichloroacetic acid (2% final concentration) was added and samples were centrifuged at 2500× *g* for 10 min. The supernatant was mixed with 0.05 M NaOH, and the amount of phenol red was measured colorimetrically at 560 nm using a microplate reader (Multiskan GO, Thermo Fisher Scientific, USA). The standard sample (zero-time control) was determined by the amount of phenol red recovered from mice sacrificed immediately after oral administration of 1.5% carboxymethylcellulose containing 0.05% phenol red. The gastric emptying rate was calculated using the following Equation: (1)$$\mathrm{Gastric}\;\mathrm{Emptying}\;\mathrm{rate}\;\left(\%\right)=\left[1–\frac{\mathrm{ABS}560\mathrm{nm}\;\mathrm{of}\;\mathrm{test}\;\mathrm{stomach}}{\mathrm{ABS}560\mathrm{nm}\;\mathrm{of}\;0\;\mathrm{time}\;\mathrm{control}\;\mathrm{stomach}}\right]\times 100$$

### 2.6. Statistical Analysis

Statistical analysis of the experimental results was performed with Student’s t-test, using the Prism program (GraphPad Prism quickCalcs, San Diego, CA, USA). Statistical significance was set at *p* < 0.05.

## 3. Results

### 3.1. Blood Glucose and Body Weight in the STZ-Induced Diabetic Mice Model

Two weeks after completion of the 3-day treatment with STZ or citric acid buffer, STZ-induced diabetic mice showed significantly higher blood glucose levels and lower body weight compared to normal controls (*p* < 0.05, Figure 2). There was no significant difference in the blood glucose concentration and the degree of body-weight loss in the diabetic group.

DA-9701 had no effect on either blood glucose level (Figure 3) or body weight (Figure 4) in STZ-induced diabetic mice.

### 3.2. GI Motility in the STZ-Induced Diabetic Mice Model

The isometric contractions of the antrum and small bowel were significantly decreased in the STZ-induced diabetic mice group compared to the normal control group (100% vs. 87.4%, Figure 5A, and 100% vs. 87.4%, Figure 5B). However, colonic motility was not significantly different between the groups (Figure 5C).

### 3.3. Effects of DA-9701 on GI Motility

The antral motility decreased in STZ-induced diabetic mice before any treatment and was increased after DA-9701 treatment in a dose-dependent manner (Figure 6). Although the antral motility in the diabetic group was improved according to the increased dosage of DA-9701, the amplitude was higher in the normal control group than in the diabetic group. However, the amplitude of antral motility was even higher in the diabetic group than in the normal control group at a dose of >100 mg/kg of DA-9701.

In both the normal control and diabetic groups, DA-9701 increased spontaneous movement of the small intestinal muscle in a dose-dependent manner, without significant differences between the two groups. Motility showed no significant improvement in either diabetic or normal control groups under 30 mg/kg of DA-9701. However, the ileal muscular amplitude was significantly improved at over 30 mg/kg of DA-9701, with an increasing linear pattern (*p* < 0.05). This improvement was more prominent in the diabetic group (Figure 7).

Baseline colonic motility showed no definite difference between the diabetic and normal control groups. Despite the similar initial contractility between the two groups, colonic motility was dramatically improved after DA-9701 treatment in the diabetic group in a dose-dependent manner. In the control group, colonic motility did not respond to treatment with 30 mg/kg of DA-9701 and improved at over 30 mg/kg of DA-9701. However, the improvement was less than that in the diabetic group (Figure 8).

### 3.4. Small Intestinal Transit and GE Rate in the STZ-Induced Diabetic Mice

The intestinal transit decreased significantly in STZ-induced diabetic mice compared to normal controls (43 ± 7% vs. 67 ± 8%, *p* < 0.05, Figure 9).

The GE rate was also significantly delayed in the diabetic group compared to that in the normal control group (43 ± 10% vs. 62 ± 12%, *p* < 0.05, Figure 10).

### 3.5. Effects of DA-9701 on Intestinal Transit

Significantly decreased intestinal transit in STZ-induced diabetic mice was improved with DA-9701 treatment, compared to placebo (3% HPMC). The intestinal transit was 67 ± 4.0%, 60 ± 4.2%, and 70 ± 6.6% in the treatment with 1 mg/kg, 3 mg/kg, and 10 mg/kg of DA-9701, and 49 ± 4.0% placebo (3% HMC) groups, respectively. In particular, the intestinal transit improved significantly with 1 mg/kg and 10 mg/kg of DA-9701 treatment compared to placebo (Figure 9).

### 3.6. Effects of DA-9701 on GE Rate

Delayed GE in STZ-induced diabetic mice improved with administration of DA-9701 in a dose-dependent manner (50 ± 11 %, 55 ± 12%, and 60 ± 11% at 1 mg/kg, 3 mg/kg, and 10 mg/kg of DA-9701, respectively). In particular, treatment with 10 mg/kg of DA-9701 significantly improved the GE rate compared to the placebo-treated diabetic group (60 ± 11% vs. 43 ± 10%, *p* < 0.05, Figure 10).

## 4. Discussion

Diabetic gastroparesis affects 20–50% of people with type 1 diabetes and 5% of patients who have been diagnosed with type 2 diabetes for more than 10 years [8]. The stomach is more susceptible to diabetic complications than the small intestine; approximately 75% of patients with diabetes have gastrointestinal symptoms, one-third of whom have stomach symptoms [9,10].

It is considered that this disease spectrum can be caused by autonomic neuropathy [3,11]. Diabetes causes loss of myelinated sympathetic trunk fibers and enlarged dystrophic axons and nerve terminals in the prevertebral sympathetic ganglia [12]. ICC loss in diabetes has been observed in human and animal models [13,14]. Reduced insulin and IGF-1 signaling may cause ICC loss, smooth muscle atrophy, and reduced stem cell factor production [15]. Another mechanism is that oxidative stress results in the loss of heme oxygenase 1 (HO1)-containing macrophages. HO1 is an enzyme expressed in macrophages and has a protective effect against oxidative stress. The main source of HO1 is the stomach muscle wall [10]. Pathological findings associated with gastric sensorimotor dysfunction can occur; these include delayed GE, gastric dysrhythmia, fundic accommodation, weakened antral pump, antroduodenal discoordination, duodenal neuromuscular dysfunction, and abnormal duodenal feedback [16,17].

This study presented several important findings associated with diabetic gastroparesis and gastroenteropathy. First, GI motility significantly decreased with site selectivity in diabetic conditions. The bowel motility power was found to be weak from the stomach to the small intestine in a diabetic model; however, colonic motility showed no significant decline. Although various factors may affect the colonic symptoms, this finding is one explanation for why diabetic patients present with various colonic symptoms, such as diarrhea, abdominal pain, and constipation. Second, GI transit and GE were appreciably delayed in STZ-induced diabetic mice compared to normal controls. Finally, DA-9701, which has been developed as a prokinetic drug, restored the decreased GI motility and delayed intestinal transit and GE in STZ-induced diabetic mice.

The present study confirmed that gastric and small intestinal motility decreased in STZ-induced diabetic mice compared to that in normal mice. However, colonic motility was not significantly decreased in STZ-induced diabetic mice. There is a lack of well-designed studies on abnormal colonic motility in patients with diabetes [18]. However, there is a pathological change in large bowel motility because lower GI symptoms, such as diarrhea or constipation, are easily noted in longstanding diabetic patients. There is some evidence of delayed colonic transit, colonic myenteric neuronal loss, [19] anorectal dysfunction due to impaired external anal sphincter function, diminished rectal sensation to distension [18], and increased oxidative stress in colonic tissues [19]. Although muscular strength in the colon was preserved in our model, there could be other structural abnormalities or early changes in another colonic site. The definite mechanism in diabetic gastroparesis should be revealed in future studies using multiple samples from the GI tract.

This study showed that DA-9701 was effective in improving GI motility, small bowel transit, and gastric emptying in STZ-induced diabetic mice. Interestingly, the changes in motility amplitude were greater in the colon than in the antral or small intestine after DA-9701 treatment. Although the baseline colonic motility in STZ-induced diabetic mice was not different from that in control mice, the response to the drug was more remarkable in the colon than in other organs. DA-9701 improved functional constipation in non-diabetic patients by accelerating colonic transit in a single-center experience [20]. However, this drug did not improve constipation symptoms in patients with Parkinson’s disease [21]. Further studies are required to determine whether DA-9701 would improve lower GI symptoms more than upper GI symptoms in diabetic patients, and furthermore, how DA-9701 increases colonic motility more than other organs in a diabetic model.

In this study, DA-9701 improved both small intestinal transit and GE in a dose-dependent manner. Many investigators have reported the effect of DA-9701 on the restoration of reduced GI transit and GE. Ramsbottom et al. [22] compared the efficacy of DA-9701 in GE with cisapride, clonidine, and apomorphine in a delayed GE model. The prokinetic effects of DA-9701 were comparable to those of cisapride (10 mg/kg) in normal animals and a delayed GE model at doses of 0.3–3 mg/kg [6]. In an in vitro study using an ICC, it was suggested that DA-9701 might affect GI motility by modulating pacemaker activity in the ICC [23,24]. DA-9701 accelerated GE, which was confirmed by a 13C-octanoic acid breath test with repeated measurements in normal mice [25]. DA-9701 improved stress-induced delayed GE, and this effect might be associated with the inhibition of stress-induced increases in plasma levels of adrenocorticotropic hormone (ACTH) and ghrelin [26]. The first multicenter, double-blind, randomized clinical trial was performed with concealed allocation, comparing the safety and efficacy of DA-9701 and itopride hydrochloride in Korea [24]. DA-9701 significantly improved both functional dyspepsia symptoms and quality of life in patients with functional dyspepsia. The efficacy of DA-9701 was not inferior to that of itopride [24]. Although DA-9701 was effective in treating functional gastric diseases, the effects of this drug in diabetic patients have not been proven yet. Thus, our results provide evidence for the use of DA-9701 in the treatment of diabetic gastroparesis and gastroenteropathy. Because of the heterogeneous symptom pattern and various causes of functional gastroparesis in diabetic patients, effective and safe doses of DA-9701 according to symptoms should be confirmed in a future study.

This study has several limitations. First, colonic motility was not significantly decreased in patients with diabetes. This might be explained by insufficient diabetic stimulation of the colon or the short duration of diabetes. Dysregulated colonic motility occurs in the advanced diabetic stage. Second, a colonic transit study of diabetes patients was not included in this study. To demonstrate the mechanism of colonic motility improvement, colonic transit or neuronal studies in colonic tissue should be performed. Further studies on the pathological findings and colonic motility function using DA-9701 will be required in diabetic animal models and clinical settings. Finally, the clinical application of DA-9701 should be added to prove its positive effect on diabetic gastroparesis. Although DA-9701 was proven to be effective in functional gastroparesis, diabetic gastroparesis is different from general functional disorders.

## 5. Conclusions

In conclusion, DA-9701 improved GI contractility without affecting blood sugar and body weight in STZ-induced diabetic mice, similar to the normal controls; in particular, it had a good effect on increasing colon motility in normal and diabetic mice. The efficacy of DA-9701 in intestinal transit and gastric emptying was similar in STZ-induced diabetic mice and normal mice. DA-9701 might be helpful in improving decreased upper and lower GI motility and various clinical symptoms in patients with progressive diabetes. Further studies are required to evaluate the effect of DA-9701 on various bowel motility dysfunctions, even in advanced diabetic patients.

## Figures and Tables

**Figure 1 jcm-10-05282-f001:**
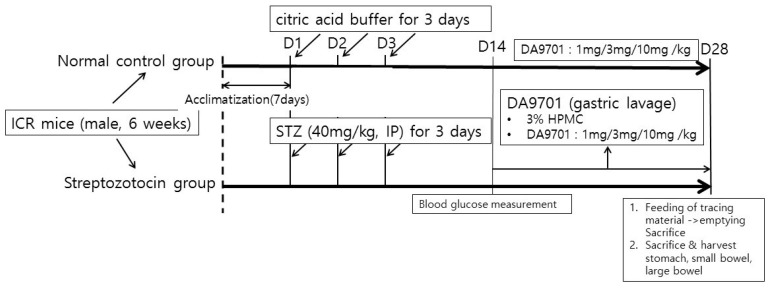
Schematic diagram of the experimental protocol.

**Figure 2 jcm-10-05282-f002:**
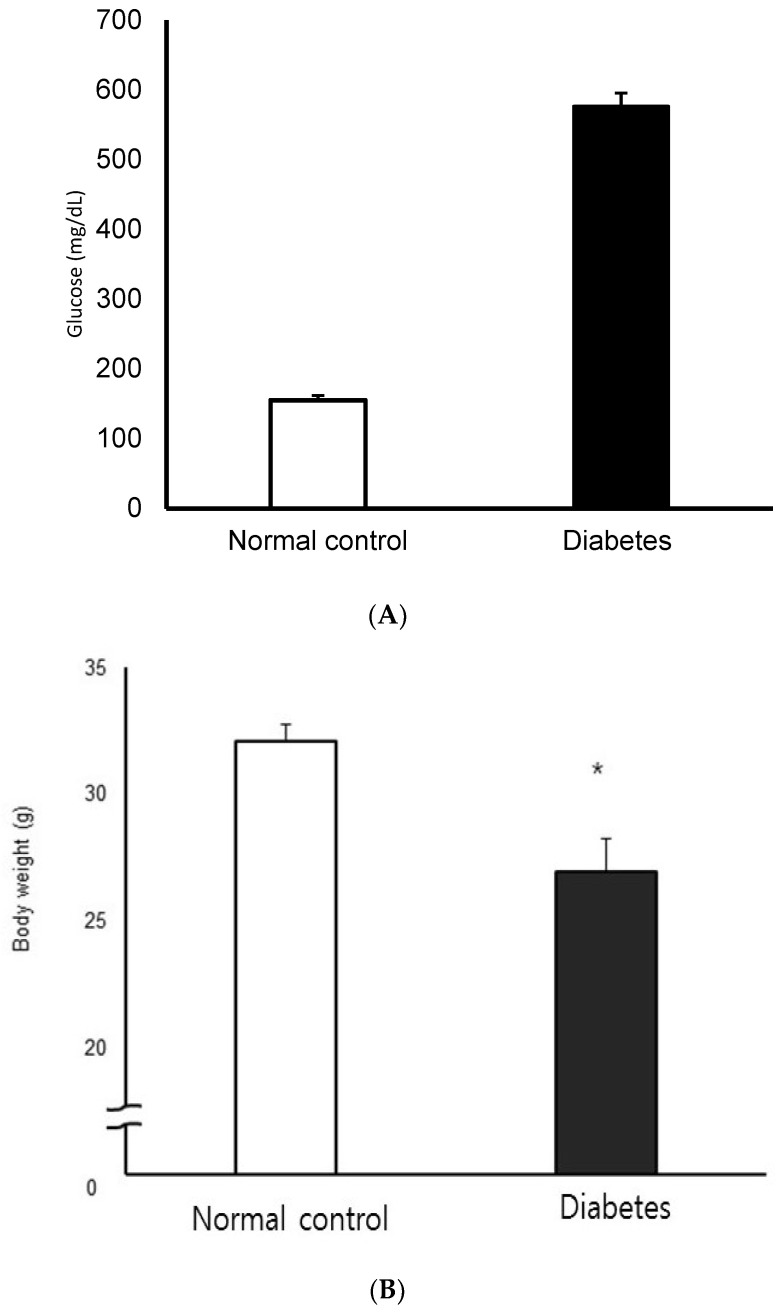
Blood glucose concentration (**A**) and body weight (**B**) in normal control and streptozotocin-induced diabetic mice. STZ-induced diabetic mice showed significantly higher blood glucose levels and lower body weight compared to normal control (n = 10 in each group) * *p* < 0.05.

**Figure 3 jcm-10-05282-f003:**
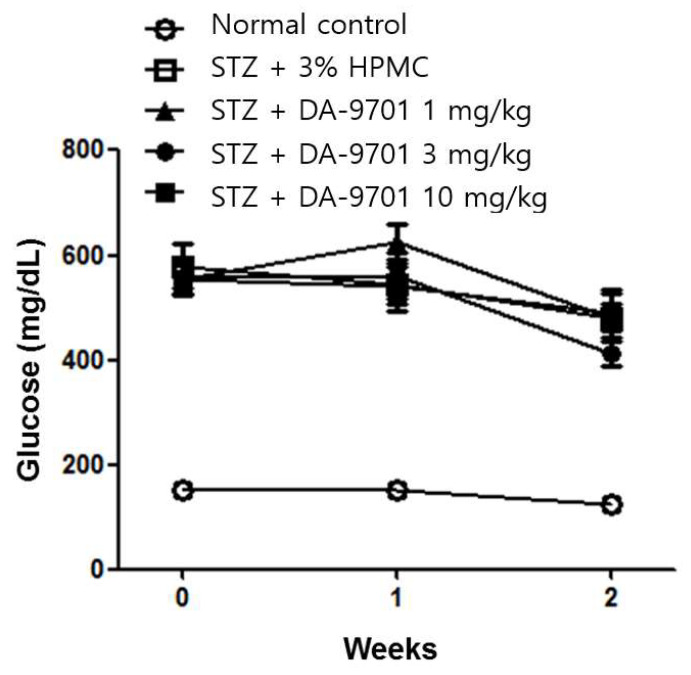
Effect of DA-9701 on blood glucose level in normal control and streptozotocin-induced diabetic mice. Streptozotocin-induced diabetic mice showed significantly higher blood glucose levels than control. DA-9701 did not affect blood glucose level in the streptozotocin-induced diabetic mice. (n = 10 in each group).

**Figure 4 jcm-10-05282-f004:**
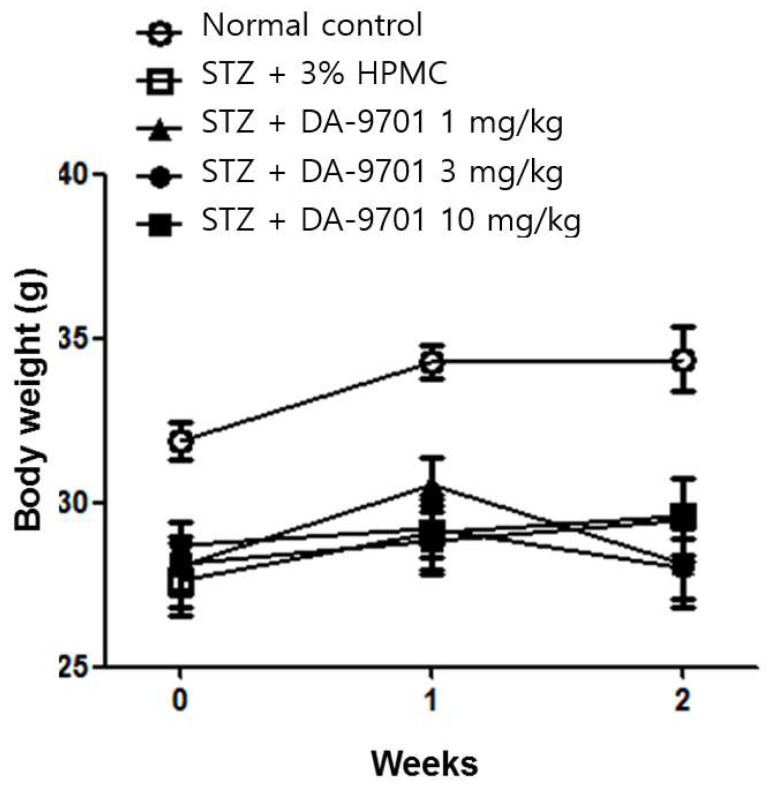
Effect of DA-9701 on body weight in normal control and streptozotocin-induced diabetic mice. STZ-induced diabetic mice had significantly lower body weight compared to control. DA-9701 had no effect on the body weight of the streptozotocin-induced diabetic mice. (n = 10 in each group).

**Figure 5 jcm-10-05282-f005:**
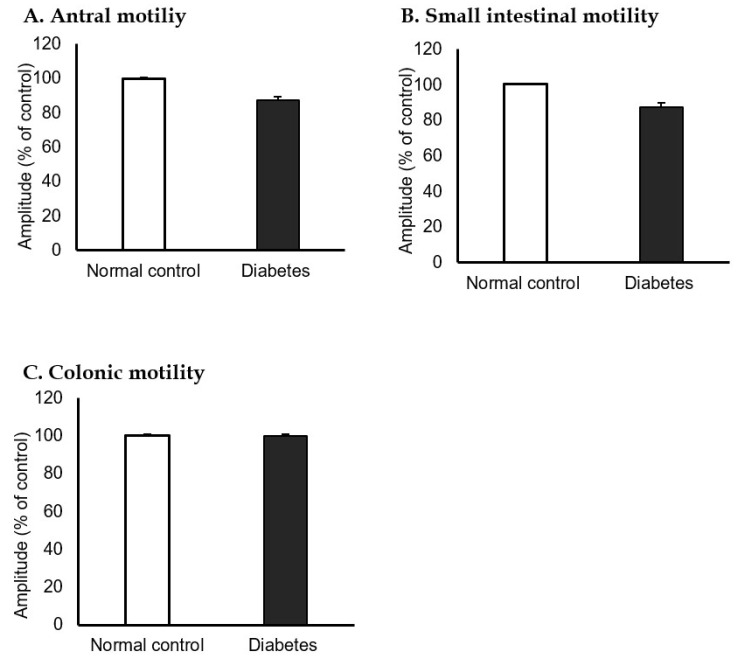
Antral, small intestinal, and colonic motility in normal control and streptozotocin-induced diabetic mice. The isometric contractions of each (**A**) antrum and (**B**) small bowel were significantly decreased in the streptozotocin-induced diabetic group compared to the normal control group (100% vs. 87.4%, 100% vs. 87.4%). (**C**) Colonic motility was not different between the normal control and diabetic group (100% vs. 100%). (n = 10 in each group).

**Figure 6 jcm-10-05282-f006:**
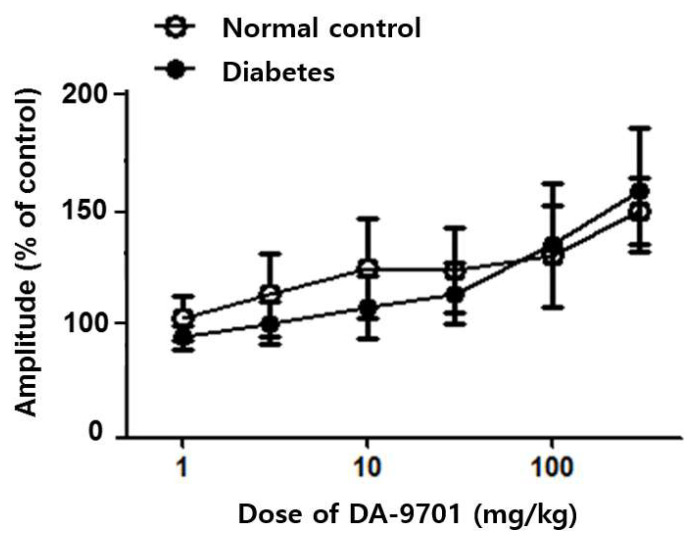
Effect of DA-9701 on antral motility in normal control and streptozotocin-induced diabetic mice. DA-9701 increased the spontaneous movement of the pyloric sinus ciliary muscle in a dose-dependent manner in both normal control and diabetic groups, without significant difference between two groups. (n = 10 in each group).

**Figure 7 jcm-10-05282-f007:**
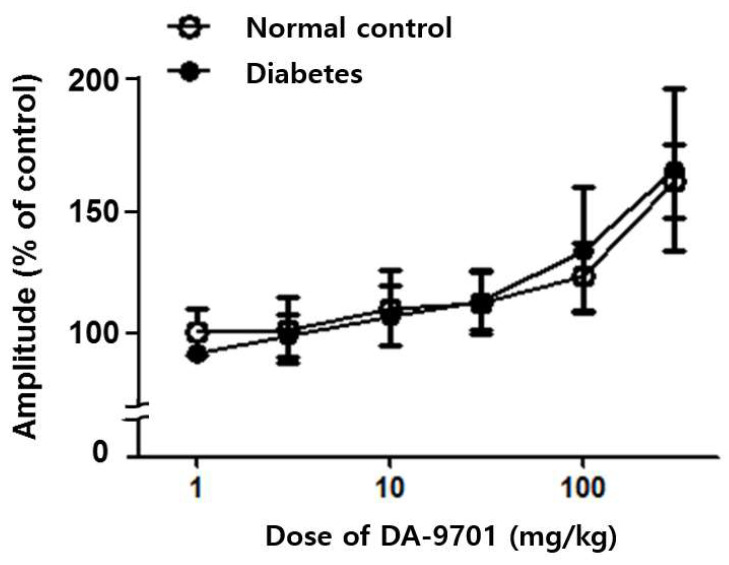
Effect of DA-9701 on small intestinal motility in normal control and streptozotocin-induced diabetic mice. In both the normal control and diabetic groups, DA-9701 increased the spontaneous movement of the small intestinal muscle in a dose-dependent manner, without significant difference between two groups. (n = 10 in each group).

**Figure 8 jcm-10-05282-f008:**
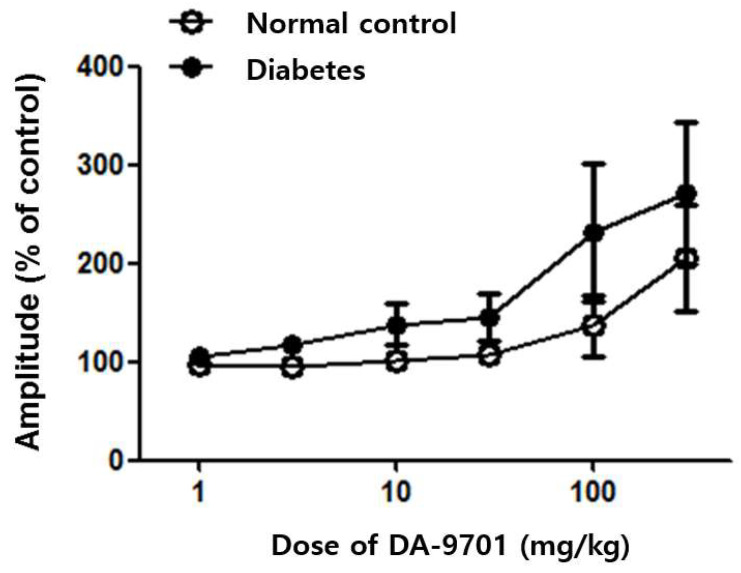
Effect of DA-9701 on colonic motility in normal control and streptozotocin-induced diabetic mice. DA-9701 administration showed a dose-dependent increase in the amplitude of spontaneous phasic contractions in the colon in normal controls and streptozotocin-induced diabetic mice, without significant difference between the two groups. (n = 10 in each group).

**Figure 9 jcm-10-05282-f009:**
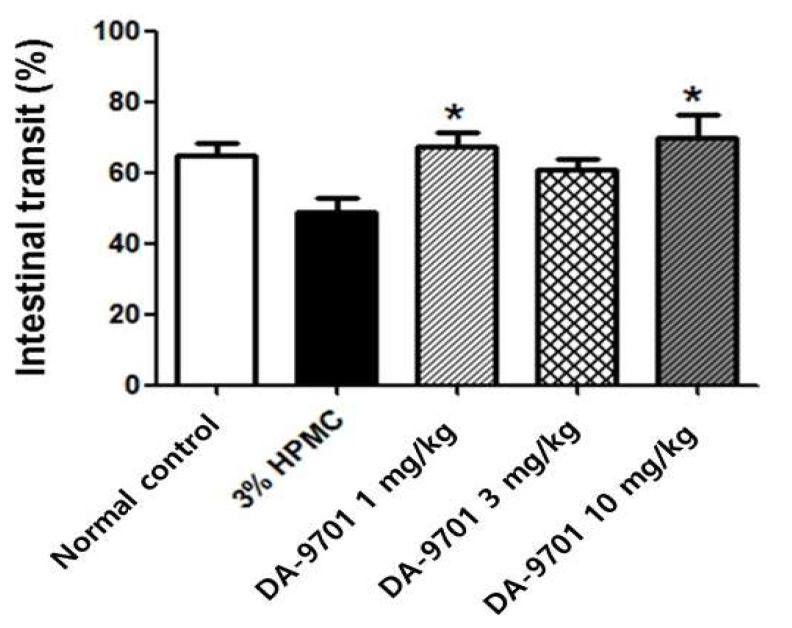
Effect of DA-9701 on intestinal transit in streptozotocin-induced diabetic mice. Intestinal transit was significantly decreased in streptozotocin-induced diabetic mice compared to normal control and improved in DA-9701 treatment group compared to placebo (3% HPMC-treated) group. The intestinal transit was 67 ± 4.0%, 60 ± 4.2%, 70 ± 6.6% in the treatment with 1 mg/kg, 3 mg/kg, and 10 mg/kg of DA-9701, and 49 ± 4.0% placebo groups, respectively. In particular, the intestinal transit improved significantly with 1 mg/Kg and 10 mg/kg of DA-9701 treatment compared to placebo (n = 10 in each group) * *p* < 0.05.

**Figure 10 jcm-10-05282-f010:**
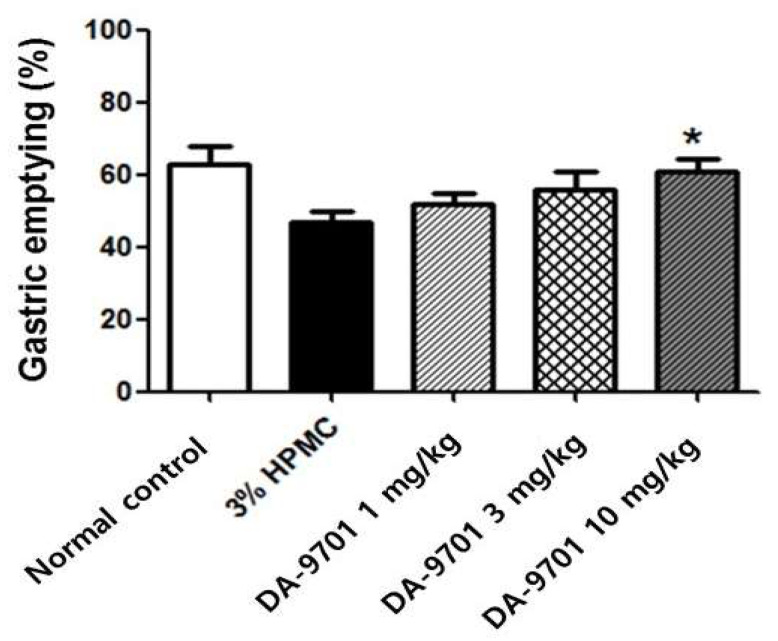
Effect of DA-9701 on gastric emptying in streptozotocin-induced diabetic mice. Gastric emptying was significantly delayed in streptozotocin-induced diabetic mice and improved with DA-9701 in a dose-dependent manner (50 ± 11 %, 55 ± 12%, and 60 ± 11% in 1 mg/kg, 3 mg/kg, and 10 mg/kg of DA-9701 treatment, respectively), compared to the 3% HPMC-treated placebo group. In particular, treatment with 10 mg/kg of DA-9701 significantly improved gastric emptying rate compared to placebo (60 ± 11% vs. 43 ± 10%, * *p* < 0.05) (n = 10 in each group).

## Data Availability

The data presented in this study are available on request from the corresponding author.

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
