# Peer review of "Effect of DA-9701 on the Gastrointestinal Motility in the Streptozotocin-Induced Diabetic Mice"

_jcm, 2021, doi:10.3390/jcm10225282_

Round 1

Reviewer 1 Report

Thank you for the opportunity to review this manuscript. This is an important and interesting area of research work.

In my opinion, the manuscript was prepared well, so I suggest accepting it in actual form.
*       The Introduction Section explains the design of the study. The Authors well justify the research topic.
*       The study was carried out without methodological errors.
*       The Descriptions of the results were correct.
*       The presented figures and table were prepared precisely and also legible.
*       The Discussion Section includes the accurate reference of the results obtained to the studies of other studies.
*       The Conclusions were well formulated.

Author Response

Thank you very much for your kind review. 

Reviewer 2 Report

Dear authors, 

I commend you on your hard work. Here are my suggestions for improvement:

-One of the major limitations on your manuscript is that I did not have a sense on how many mice were use used in your study. It explained how the control and the STZ treated mice were separated into groups, but no clear numbers were provided so that the reader can assess the statistical power of your study. My recommendation is that clear number be provided for each study group. 

-I apologize, but I am not as well familiar with one part of your study method, so my questions is why you chose to administer STZ within a citric acid buffer? Second I would specify if the STZ dosage numbers that are given in mg/Kg, are for the STZ alone or STZ citric acid buffer solution? In other words, I would specify if the dosage number given in the manuscript are for STZ alone or the STZ citric based solutions?

-Your charts are very helpful, and you properly explain them in the written portion underneath the figure. However, I would consider labelling them as "Antrum motility", "Small intestinal motility", etc. This is because it facilitates the reader to locate certain chart or tables without having to continuously read the lower figure explanation. 

-On page 15 and 17, you have two sections titled as "Intestinal transit and GE rate in the STZ-induced diabetic mice" and "Effects of DA-9701 on intestinal transit", followed by an explanatory paragraph. However, I am not sure if you are referring to small or large intestine? Or if you are including both?

-In your Discussion, you write "however, colonic motility showed no significant decline. This finding could explain why diabetic patients present with various colonic symptoms, such as diarrhea, abdominal pain, .....". This statement is confusing because it is directly implying that the lack of colonic motility decline results in the symptoms given, which is not necessarily true. Other factors may result in those symptoms. I would further clarify this statement to avoid confusion by the reader.  

-On page 20, first paragraph, 2nd sentence, you write the acronym "PD", but I could not find the full word that is equivalent to this acronym. I assume it means peptic disease, but I recommend that the author use the full word, since that acronym is hardly used elsewhere in the manuscript.  

-On page 21, conclusion, 1st sentence, you mention that "DA-9701 improved contractility.........in particular, it had good effect on increasing colon motility in normal and diabetic mice." This statement is confusing to me, since it implies that the normal control mice were also given DA-9701, and from your method section, I thought that only STZ treated mice had this intervention. Please further clarify. 

Thank you for giving us the privilege of reading your manuscript.   
